ions

# Centripetal nuclear shape fluctuations associate with chromatin condensation in early prophase

Viola Introini[1,2,5], Gururaj Rao Kidiyoor [3,5], Giancarlo Porcella[3,5], Pietro Cicuta [1] &
Marco Cosentino Lagomarsino [3,4 ✉]

The nucleus plays a central role in several key cellular processes, including chromosome organisation, DNA replication and gene transcription. Recent work suggests an association between nuclear mechanics and cell-cycle progression, but many aspects of this connection remain unexplored. Here, by monitoring nuclear shape fluctuations at different cell cycle stages, we uncover increasing inward fluctuations in late G2 and in early prophase, which are initially transient, but develop into instabilities when approaching the nuclear-envelope breakdown. We demonstrate that such deformations correlate with chromatin condensation by perturbing both the chromatin and the cytoskeletal structures. We propose that the contrasting forces between an extensile stress and centripetal pulling from chromatin condensation could mechanically link chromosome condensation with nuclear-envelope breakdown, two main nuclear processes occurring during mitosis.

[1] Cavendish Laboratory, University of Cambridge, J.J. Thomson Avenue, Cambridge CB3 0HE, UK. [2] Cambridge Institute for Medical Research, University of Cambridge, Cambridge Biomedical Campus Keith Peters Building, Hills Rd, Cambridge CB2 0XY, UK. [3] IFOM, FIRC Institute of Molecular Oncology, Via Adamello 16, Milan 20139, Italy. [4] Dipartimento di Fisica, Università degli Studi di Milano and I.N.F.N., Via Celoria 16, Milan 20133, Italy. [5] These authors contributed equally: Viola Introini, Gururaj Rao Kidiyoor, Giancarlo Porcella. ✉email: marco.cosentino-lagomarsino@ifom.eu

Shape fluctuations of vesicles in vitro (also known as flickering) are driven by thermal motion, to which membranes respond passively. Specifically, these transient shape fluctuations can be interpreted in terms of equilibrium states[1], and their measurement by time-lapse microscopy provides a powerful biophysical tool to characterize mechanical parameters such as bending modulus, tension and viscosity[2]. These tools and the assumption of thermal equilibrium are valid for some simple biological models that do not experience action of molecular motors and other active forces, such as erythrocytes[3,4]. On the other hand, nucleated eukaryotic cells are able to both sense and generate mechanical forces when interacting with their surroundings to maintain cellular homeostasis[5].

In more complex living systems, chemical energy from ATP is turned into mechanical forces, e.g. by molecular motors, and these forces add to the thermal forces to induce fluctuations of cells and cellular compartments[5–7]. In this scenario, the extent of nuclear shape dynamic change is related to both passive relaxation and to the active (non-equilibrium) mechanics from forces due to the cytoskeleton or other intra and exo-nuclear force-generating processes. Teasing out these two contributions is very difficult and has been achieved only in a few cases[6,7]. It requires performing deformation assays under the presence of a known external force, and comparing the outcome to the spontaneous deformations.

Abundant evidence shows that non-equilibrium processes can drive membrane flickering to more complex behavior than predicted by thermodynamics equilibrium, for example causing a breakdown of the "fluctuation-dissipation" theorem valid at equilibrium[7], which links the decay of spontaneous fluctuations to the response to external perturbations. In such conditions, monitoring shape fluctuations is still useful, but extraction of precise biophysical parameters such as tension or stiffness are more difficult, and one can generally refer to "effective" (or "apparent") tension and bending moduli as a complex byproduct of passive membrane properties and the result of active driving forces. In such conditions the equilibrium model may still be a useful guide, for example, allowing to compare the relative amplitudes of fluctuations at different wavelengths. In many cases, the active fluctuations can be reduced to the standard model with an "effective temperature", by which the active forces simply increase the noise level with respect to the thermal motion[8].

Previous research has shown that the cell nucleus undergoes rapid shape fluctuations during the cell cycle, as observed by analyzing nuclear envelope (NE) flickering at millisecond timescales[9]. The NE is a phospholipid bilayer that encloses genetic material and is contiguous with the endoplasmic reticulum. The lamina filament network underlying the NE provides structural support and connects chromatin to the cytoskeleton via the LINC complex[10]. Shape fluctuations of the NE are likely actively generated both internally by nuclear components (as shown by an increase of undulations upon inhibition of transcription) and externally by the cytoskeleton[9]. The osmotic force balance is the primary component of the forces felt by the nuclear surface[11,12]. F-actin fibers were shown to exert a pressure gradient on the nucleus, thereby generating nuclear fluctuations in the process of nuclear positioning in oocytes[13]. Additionally, microtubules are major drivers of nuclear envelope breakdown in mitosis and have also been reported to drive stem cell differentiation by force transmission to the nucleus[14].

Recent studies show a link between nuclear mechanics and cell cycle progression in cancer cells and epithelia, linking nuclear tension to the G1/S transition[15–17]. The nucleus has also been reported to act as a "ruler" sensing mechanical forces and regulating the cell response by cPLA2-mediated actomyosin contractility[18]. Intriguingly, cells in G2 appear to have a larger such ruler, hence requiring less confinement than G1 cells to trigger the contractile response. More recently, Dantas and coworkers reported a direct role of nuclear mechanotransduction in regulating cell commitment to mitosis[19]. The authors identify a threshold for nuclear envelope tension at the G2/M boundary above which cyclin B1 internalizes to the nucleus to initiate cell division. An increasing body of evidence thus supports the emerging role of the nucleus mechanical properties as important regulators of cell-cycle transitions. However, less is known about the chromatin-specific contributions to nuclear shape fluctuations, particularly in mitosis, were chromatin and cytoskeletal dynamics must be finely tuned to achieve cell division. Here, by high-speed confocal live imaging of nuclear envelope, we analyze its shape dynamics and fluctuations throughout the cell cycle. By drug perturbation of the cytoskeleton and of chromatin rearrangement, we isolate the role of chromatin in generating these fluctuations, in particular during condensation in mitosis. Our results support a role of chromatin in exerting mechanical forces on the NE during the series of events leading to cell division.

## Results

**Nuclear shape fluctuations vary with cell cycle progression**. We first tested whether our cells showed a similar cell-cycle dependency of the NE flickering as observed by Chu et al. To do so, we used a GFP-tagged version of the Emerin protein to mark the NE and minimize the adverse effects of labeling on nuclear behavior, since it is well known that Lamin over-expression can directly impact mechanical properties of the nucleus[20]. Analyzing the cell cycle duration of the HeLa cells by performing time-lapse videos (see Supplementary Information) showed that on average, in our growth conditions, HeLa cells spent about 16.5 h in interphase and about 1.75 h undergoing mitosis (Fig. S1a), consistent with previous reports[21]. The nuclear area increased throughout interphase, slightly decreasing with entry to prophase, when the chromatin condensation was apparent. (Fig. S1b). Cells were arrested at the G1/S transition (by a double thymidine block) or at the G2/M transition (by CDK1 inhibition) and then released to follow them through cell cycle progression by monitoring the cells every 3 h from release (Fig. 1a). The approximate cell cycle stages were assigned after counting the elapsed time after their release from arrest, using reported timings of HeLa cell cycle phases (G1 7–8 h, S 7–8 h, and G2 2–3 h)[22] as a reference (Fig. 1a).

To analyze NE flickering along the cell cycle, we monitored the average fluctuation spectra at different stages of cell growth. This is defined as the amplitude of fluctuations, calculated by the deviation of the instantaneous contour from the average contour for all recorded frames, plotted as a function of wave vector (inverse wavelength of the projected shape deformation) (Fig. 1b)[23]. Nuclear fluctuations decrease by about threefold from early G1 (~2–3 h after mitosis) to late S phase (~12 h post mitosis or 6 h post G1/S release, see "Methods" section and Video S1), which is in line with data shown by Chu and coworkers after 13 h of release from mitosis arrest[9]. However, we also report a notable increase of these fluctuations already in late G2 (CDK1 arrested cells) and early prophase (Video S2 and Fig. 1b). During later stages of prophase, as the cells start to round up, these fluctuations evolve into dramatic deformations, acquiring increasing instability, which culminates with the NEBD.

Although the system is out of equilibrium, if we assume that active forces play the role of an increased "effective temperature" then it is possible to use the standard model for fluctuations to extract effective biophysical parameters[9]. Importantly, these measured effective parameters are not the same as the biophysical

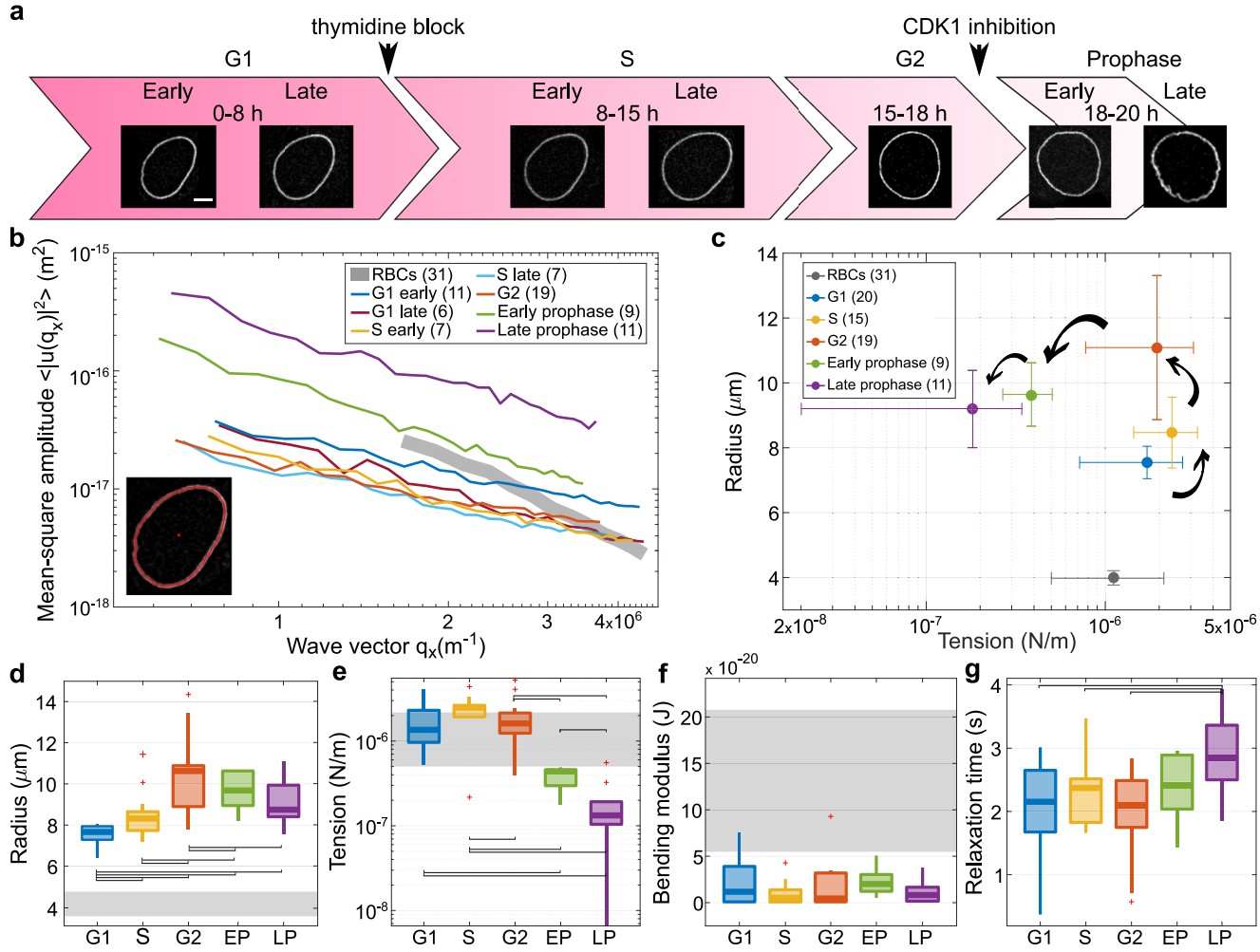

**Fig. 1 Shape fluctuations of HeLa cell nuclei are cell-cycle dependent and increase towards early mitosis. a** Snapshots of a representative nucleus at 7 time points from the start at early G1 phase, throughout S, G2 and prophase (scale bar 5 μm). Arrowheads indicate the reference time-points to determine cell cycle phase (see "Methods" section). **b** Average spectra of wave vector-dependent fluctuation amplitudes (modes 6–34) for cells at different stages in the cell cycle. The number of nuclei considered for each cell-cycle stage are reported in the legend in brackets. The fluctuation amplitude $\langle u_q^2 \rangle$ exhibits a decrease with increasing time from G1 until G2, where the fluctuations are reduced by about three times. Instead, active nuclear fluctuations during mitosis become four times higher in early mitosis (green line) and 10 times in late mitosis (purple line). Inset: contour detection of NE (red line) with fluorescent label Emerin. The initial manual selection of the center (red dot) and an initial point on the NE define the annular region containing the cell boundary used in image analysis. **c** Effective tension *vs* radius scatterplot shows clusters from different cell-cycle stages forming an open counterclockwise trajectory. **d**–**f** Box plots of shape-fluctuation parameters throughout the cell cycle. The data show no significant changes (*p*-value > 0.05) in effective bending modulus across the cell cycle, while effective tension increases significantly during S phase and decreases up to one order of magnitude during mitosis. The cell radius increases from the starting point in G1 until G2 and then does not change much. **g** the characteristic relaxation time for mode 3 becomes longer during mitosis. Gray bands and markers represent RBC fluctuation parameters. *P* values are reported in Table S2 and were calculated using the two-sample *t*-test; significant relations are highlighted with brackets.

ones, but a byproduct of constitutive parameters and the action of active forces. We will refer to them as "effective tension" and "effective bending modulus" in the following, and explicitly discuss their interpretation whenever necessary.

Adding to previous work, we introduce two important technical improvements. First, we account for the projection of fluctuations on the equator in the measured shape deformations[3,23,24], which were neglected in the Chu et al. study and lead to erroneous *q* dependencies. Second, we consider spectra as a function of wave vector rather than wave number, in order not to average together fluctuations of different wavelength from nuclei of different size. We obtain the average square displacement $\langle u_{q_x}^2 \rangle$, where the wave vector of the projected equatorial profile is $q_x = 2\pi n / L$, $L$ is the length of the profile and $n = 0, 1, 2, \ldots$ the modes[3]. Effective bending modulus and tension

are then obtained by a fit of the spectra with the formula

$$\langle u_{q_x}^2 \rangle = \frac{k_B T}{2\sigma} \left[ \frac{1}{q_x} - \frac{1}{\sqrt{\frac{\sigma}{\kappa} + q_x^2}} \right], \quad (1)$$

where $\sigma$ is an effective tension (measuring, in the passive case, mechanical response to extensile stress) and $\kappa$ is an effective bending modulus (measuring response to curvature in the passive case). $k_B$ is Boltzmann's constant, and $T$ is the absolute temperature. By taking $T$ as the physical temperature, we interpret any non-equilibrium effect within the parameters $\sigma$ and $\kappa$. This seems justified here because the mode-dependence of the data is consistent with the equilibrium model. Considering an active surface like the NE with this model, $\sigma$ can be interpreted as the resistance of the surface to change total area, in response to

active and thermal forces, and $\kappa$ as the total energy necessary to bend and ruffle the surface[8,25].

Equation (1) has limiting behaviors $\langle u_{q_x}^2 \rangle \sim 1/q_x$ for modes dominated by effective membrane tension, and $\langle u_{q_x}^2 \rangle \sim 1/q_x^3$ for modes dominated by effective bending rigidity. For the range of modes considered in our analysis, the nuclear fluctuations are mainly affected by effective tension, as the mean-square amplitude spectrum is dominated by the $1/q_x$ trend[4,23].

It is instructive to plot nuclear radius versus effective tension in the different cell-cycle stages (Fig. 1c). Effective tension initially increases with radius, as would be expected for an inflated passive membrane. However, this trend is inverted starting from the late G2 phase, so that radius keeps increasing while effective tension is reduced. The physical properties of the lamina may change considerably in this part of the cell cycle due to Lamin phosphorylation[26,27], but an increase in the active forces could also concomitantly drive nuclear shape fluctuations. Hence the decrease in effective tension in late G2 and prophase could be due to a drop in physical tension, and/or an effect of active forces. Overall, Fig. 1c shows how the nucleus of a cycling cell follows a counterclockwise trajectory in the effective tension-radius plane, which starts at nucleus birth and culminates in NE breakdown at late prophase. A recent study has shown that an increase of NE tension, as marked by cPLA2 translocation from the nucleoplasm, peaks at mitotic onset before NEBD[19]. Here we show that the change in physical properties of the nucleus and emerging active forces lead to a following decrease in effective tension, resulting in a less mechanically stable NE, more prone to deformation. The changes in radius and effective tension across phases of cell cycle are statistically significant (Fig. 1d–f), while effective bending rigidity, remains fairly constant. We verified which of the average trends of the parameters were robust in single-cell trajectories by looking at cells that were imaged over multiple time-points covering more than one stage of the cell cycle (Fig. S2).

From our measurements, it is possible to monitor the relaxation time scales of the dominant deformation modes. For a passive membrane, the modes decay exponentially, and their relaxation time scale is the ratio between the modulus driving the relaxation and the viscosity. When the modulus is determined from a passive spectrum, e.g. the tension, then one can use it to determine the viscosity. However, for an active surface such as the NE, the time scales reflect active dynamics, and the decay of modes can become very complex. We considered the relaxation time $\tau$ of mode 3, where we found that no complex behavior appears and the decay is a simple exponential (see Fig. S3) associates with longer relaxation times (Fig. 1g). This can be interpreted as a signature of active fluctuations/deformations from nuclear or cytoplasmic pulling or pushing elements, visible in the movies during prophase, which could trigger different characteristic times (due to the dynamics of the active elements) than those of passive relaxation.

Late prophase is associated with longer relaxation time (Fig. 1g). This can be interpreted as the signature of active fluctuations/deformations from active nuclear or cytoplasmic pulling or pushing elements, observed in the movies during prophase. These active fluctuations trigger different characteristic times (due to the dynamics of the active elements) than those of passive relaxation.

Red blood cell (RBC) fluctuations have been extensively studied, representing a simpler well-understood system, yet with some common biophysical properties in common with cell nuclei (e.g., being supported by cytoskeletal elements)[3,4]. Hence, we decided that it could be instructive to use them as a reference, and we compared the behavior of HeLa cell nuclei with those of RBC (gray bands in Fig. 1). HeLa nuclei have in general larger dimensions, a longer relaxation time, and smaller effective bending modulus, but their effective tension is similar to RBC if we exclude the dramatic changes occurring for nuclei at prophase. The mean and SEM of nuclear biophysical properties from HeLa cells at different stages of the cell cycle are reported in Table S1, and $P$-values in Table S2.

**Calyculin A treatment recapitulates the behavior of nuclear shape fluctuations during prophase**. Mitosis is a complex process, requiring considerable structural rearrangements and coordinated signaling cascades to ensure error-free cell division[28]. Several studies have addressed the biochemical aspects of this process[29], but less is known about the many mechanical requirements accompanying these processes. In a seminal study, Beaudouin and colleagues have shown how microtubules spanning from centrosomes exert tension-generating forces on the envelope, ultimately leading to its breakdown by tearing[30]. Together with the cytoskeleton, the central component of cell division is chromatin, which undergoes important rearrangements culminating in the proper division of the genetic material in two daughter cells. Chromatin was shown to actively sense and respond to mechanical stimuli, modifying the nucleus elastic properties to counteract mounting mechanical stress[31].

Following these considerations, we analyzed the role of chromatin in determining nuclear shape fluctuations at the onset of mitosis. To uncouple the effect of chromatin condensation from biochemical modifications inherent to mitosis, we treated cells with calyculin A, a drug inducing rapid premature chromatin condensation in cells independent of their cell cycle stage[32,33]. Shape fluctuations of calyculin-A treated nuclei were recorded and compared with the fluctuations of the same nuclei prior to drug treatment (Video S3). To address the respective contributions of chromatin and the cytoskeleton in early mitotic fluctuations, we treated cells arrested in G2/M with latrunculin A, an actin depolymerizing agent, then followed their NE dynamics (Video S4). Figure 2a shows the effect of short and long treatment with the two drugs on nuclear tension and radius. Short exposure (20 min) to both calyculin A and latrunculin A reduces nuclei effective tension in a similar fashion to what occurs in late G2 and early prophase cells (G2 phase serves here as the control). A longer (50 min) exposure of cells to latrunculin A produces no change in the nuclear radius and effective tension. On the other hand, prolonged treatment with calyculin A leads a subpopulation of cells to a further reduction of nuclear radius and an additional decrease in effective tension, resembling the behavior observed in late prophase. This is readily confirmed when comparing the radii and effective tension of interphase and mitotic nuclei with those of nuclei before and after treatment with calyculin A and latrunculin A (Fig. 2b, c).

Additional analysis of calyculin A treatment reveals that ~10–15 min after treatment, nuclei start showing shape fluctuations similar to early-prophase nuclei (henceforth termed "early calyculin A") without major deformation of the nuclear shape. These fluctuations then evolve into widespread invaginations as the cells start rounding up (see Video S3). This condition resembles the late stage of prophase, henceforth termed "late calyculin A", showing distorted nuclear shape as well as reduced nuclear radius. These late-stage profound deformations also closely resemble those observed concurrently with NEBD[30]. Nuclei subject to the late calyculin A effect show a drastic reduction in radius. Sketches of nuclei in Fig. 2b report typical changes in nuclear shapes after treatment, further characterized in Fig. 3a. Relaxation time of mode 3 increases upon calyculin treatment, similarly to prophase cells, while it is not affected by latrunculin A (Fig. 2d).

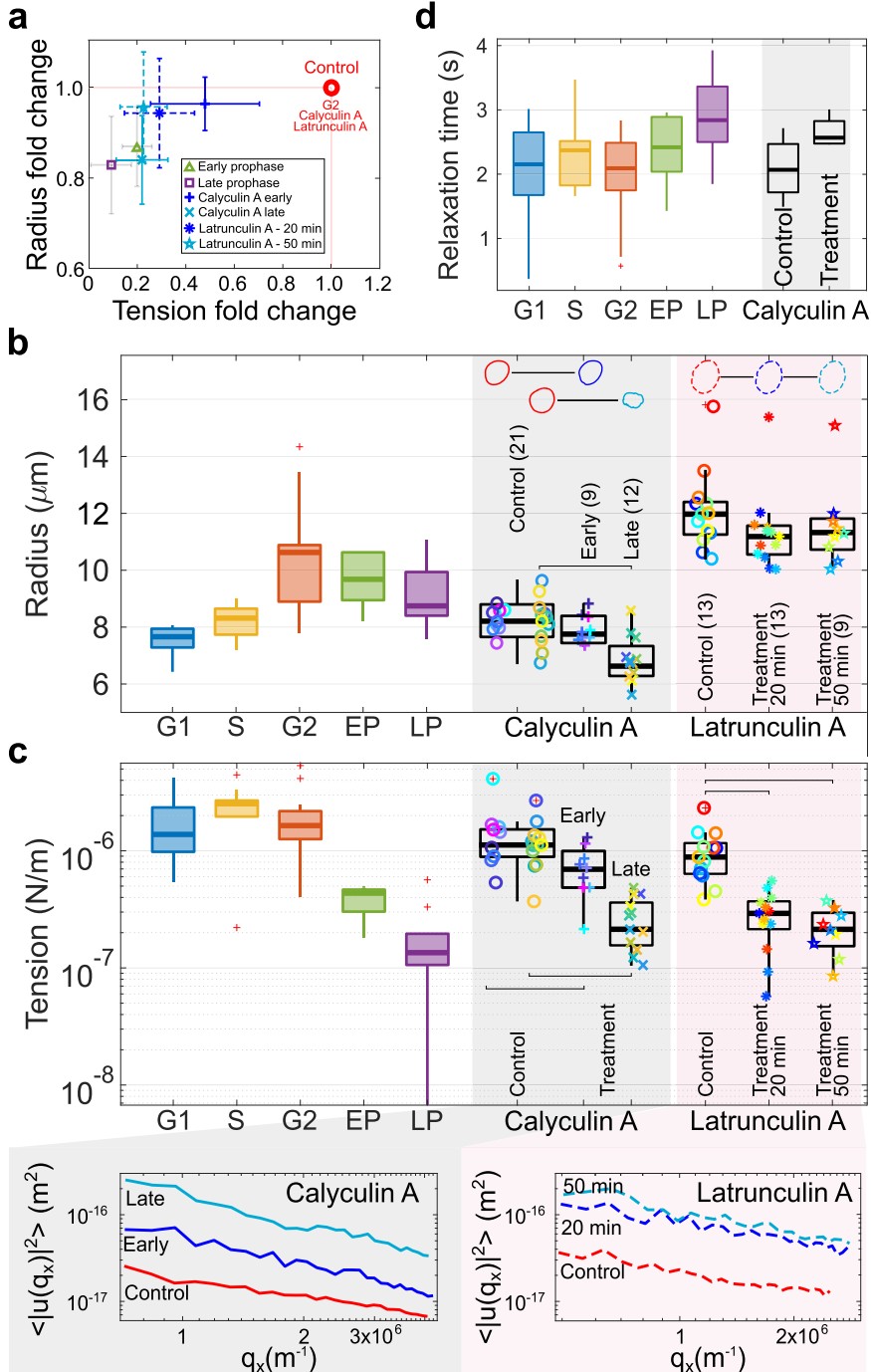

**Fig. 2 Calyculin A and latrunculin A treatments recapitulate the joint radius/effective tension changes found in prophase. a** Radius–tension change after treatments compared to the control phase G2, and early and late prophase for cycling cells. Calyculin A causes a reduction of both radius and effective tension, while latrunculin A decreases effective tension to a lower value, which remains constant with treatment time (early, 20 min vs late, 50 min). **b**, **c** Details of radius and tension of the same cells before and after both treatments, compared to the values throughout the cell cycle. Shape changes of representative nuclei are highlighted in panel **b**. The insets below panel (**c**) report the respective averaged fluctuation spectra. **d** Relaxation time of mode 3 slightly increases after calyculin treatment, as for prophase cells. Number of cells: calyculin early (13) and late (16), latrunculin 20 min (13) and 50 min (18). *P* values are reported in Table S2 and were calculated using the two-sample *t*-test; significant relations are highlighted with brackets.

Collectively, these observations suggest that calyculin A treatment (as opposed to latrunculin A treatment) recapitulates the behavior of mitotic nuclei close to NE breakdown. This leads us to hypothesize that these shape fluctuations may come to mechanical forces exerted by chromatin on the NE. Since Lamin phosphorylation could also affect NE properties and their response to forces in this stage, to rule out this possibility we assessed Lamin phosphorylation levels following calyculin A treatment, confirming that no visible change is observed (Fig. S4). Since calyculin A is also known to activate myosin-2 mediated contractility[34], we checked whether the increased centripetal invaginations following treatment could be a byproduct of increased actomyosin contractility. We performed a double chemical perturbation with calyculin A and blebbistatin (a

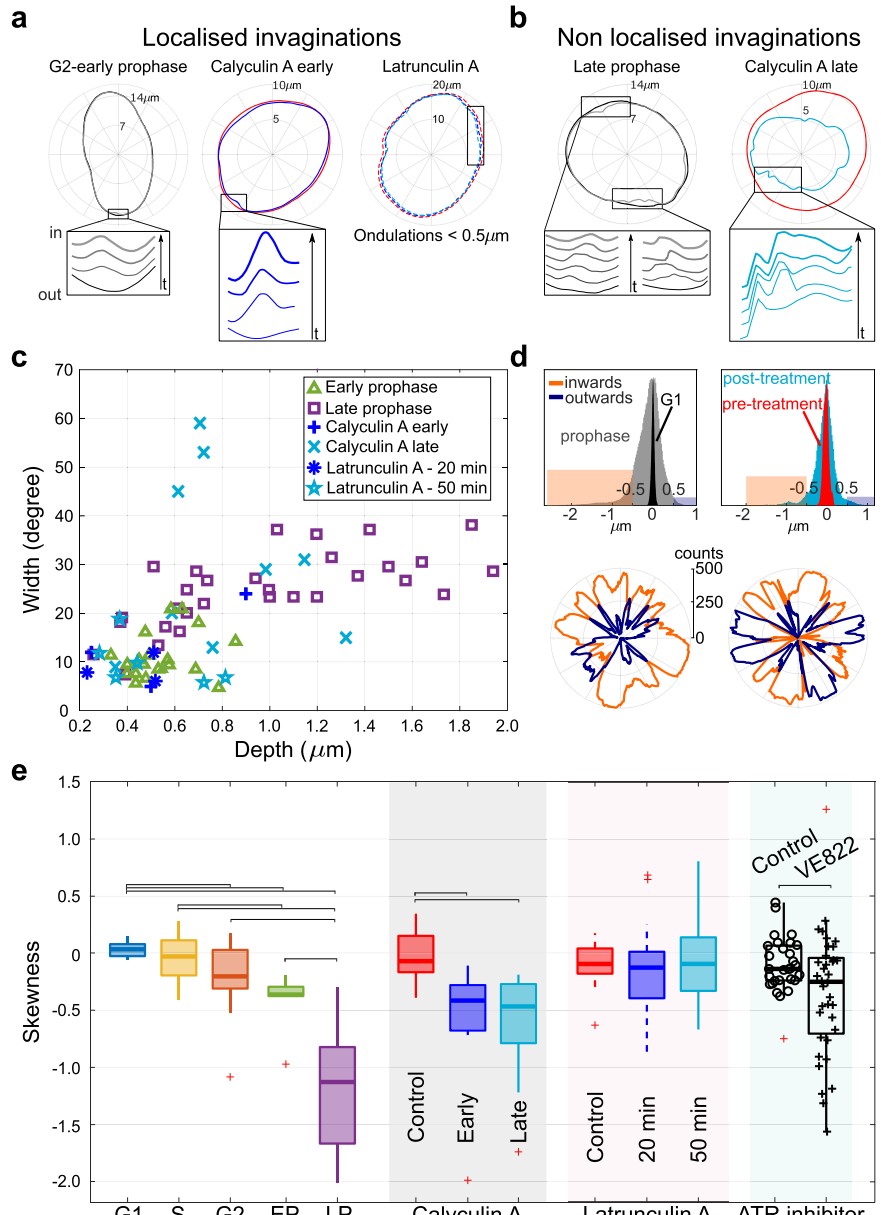

**Fig. 3 Late-G2 and prophase deformations are dominated by inwards invaginations, compatible with the action of centripetal pinning forces.**
**a** Representative examples of the localized inward invaginations that emerge in early prophase and in early stages of calyculin A treatment, but are not found with latrunculin A treatment. **b** Examples of the non-localized invaginations observed in late prophase and later stages of calyculin A treatment. The insets in panel a and b illustrate the dynamics by snapshots at equal time lags (Videos S2–S4), **c** Invagination width at the maximal deformation increases by 2-3 fold in late prophase while depth can increase up to 10 fold. Invaginations from early and late phases of calyculin A treatment resemble the ones in prophase, while latrunculin A treatment has mild effects on the invaginations (they remain within <1 μm in depth and 25 degrees in width). **d** The histograms (top) as well as polar plots (bottom) of signed shape fluctuations show the bias towards inward motion of prophase and calyculin A late nuclei (orange are inward and blue outward fluctuations). Histograms count all contour angles for 500 frames; inward fluctuations were defined as negative deformations <−0.5 μm (orange band), and outward fluctuations as positive deformations >0.5 μm (blue band). **e** Boxplots of the skewness of the signed shape fluctuation histograms (panel **d**) over cell-cycle stages and upon drug treatment. The centripetal asymmetry increases during prophase and calyculin A treatment, while latrunculin A does not affect it. ATR inhibitor VE822 increases the events with negative skewness. Number of cells for control (29) and VE822 condition (38). *P*-values are reported in Table S2 and were calculated using the two-sample *t*-test; significant relations are highlighted with brackets.

myosin inhibitor). Nuclear fluctuations, as well as their radius and effective tension, replicate nuclear features observed after treatment with calyculin A alone. Treatment with blebbistatin alone also did not affect the dominance of inwards vs outwards fluctuations, and, coherently with previous reports[9], increased effective tension (Video S5 and Fig. S5). Finally, since blebbistatin is known to be inactivated in blue light, we confirmed the result with Y27632, a rho kinase inhibitor that decreases actomyosin contractility and is not affected by illumination (Fig. S6), thus showing the same phenotype as blebbistatin treatment and as observed by Chu and coworkers[9]. These data support the interpretation that calyculin-induced nuclear shape deformations are not due to increased actomyosin contractility. The mean and SEM of nuclear biophysical properties from cells upon calyculin A and latrunculin A perturbations are reported in Table S3 together with their statistically different p-values (Table S2).

To further test the hypothesis that chromatin states can directly affect nuclear envelope dynamics, we also treated cells with Trichostatin A (TSA), a pan-histone deacetylase inhibitor, as previously described in ref. [35]. We reasoned that looser chromatin resulting from TSA treatment would lead to a decrease in the number of fluctuations observed at the NE. Indeed, TSA treatment results in a significantly reduced number of transient invaginations compared to control cells, as opposed to highly fluctuating nuclei in calyculin-A treated cells (Fig. S6). These findings suggest that chromatin relaxation translates to a lower force transmission to the NE, ultimately resulting in a lower number of transient fluctuations.

In conclusion, our data suggest that chromatin plays a central role in the generation of forces resulting in membrane fluctuations at the early entry into M phase. These can be distinguished from the profound, later occurring invaginations corresponding to onset of NEBD[30].

**Prophase fluctuations are invaginations mediated by centripetal pinning forces.** We noticed that most of the transient deformations contributing to the decrease in effective tension from G2 to mitosis had two specific properties: (i) they were localized in one region of the observed profiles and (ii) they looked like the tip of the deformation pointed towards the inner side of the nucleus (Video S2). During late prophase, the deformations became more widespread, and increased dramatically with their asymmetry towards the center of the nucleus. These likely reflect the previously observed spindle-generated forces resulting from centrosome positioning[30].

As the cells progressed towards NE breakdown, we observed that the inward deformations became increasingly long-lived and less localized, as increasingly larger patches of the lamina appeared to be displaced centripetally. Eventually, these deformations became unstable, and instead of being restored to an equilibrium shape, they developed into the deformations leading to NE breakdown. Figure 3a, b confirm this behavior, which was also found in early and late stage calyculin-A treated cells. Conversely, latrunculin A treatment does not cause invaginations, although reducing nuclear effective tension. During late prophase and late-stage calyculin A-treated nuclei, invaginations become wider and deeper (Fig. 3c). Invagination width at the maximal deformation increases by 2–3 fold in late prophase, and depth up to 10 fold. Nuclear invaginations from early and late phases of calyculin A treatment follow the trend of invaginations progressing through mitotis in untreated cells, as opposed to latrunculin A treatment, where invaginations remain within <1 μm in depth and 25° in width. Depth was calculated as the difference between the steady-state contour and the minimum of the invagination, and the width by the points corresponding to 10% of the depth. To characterize such inward deformations, we considered the distribution of signed shape fluctuations, defined as the integrated difference between the profile and a reference profile calculated as the average shape of ten frames before the invagination developed (Fig. 3d, e). Inward invaginations (<−0.5 μm, orange band in Fig. 3d) are prevalent with respect to outwards (>0.5 μm, blue band), shown both in the histograms and relative polar plots. Distributions of fluctuations for all frames and angles are wider for prophase cells (gray histogram in Fig. 3d) and post-calyculin A treatment (cyan), with respect to interphase (G1 and pre-treatment respectively black and red histograms). Polar plots represent the total number of frames of inward and outward fluctuations at every angle.

To quantify the behavior of inwards vs outwards deformations, Fig. 3e reports the skewness of their distribution as a summary statistics. A negative skewness corresponds to enrichment in

inward deformations. The results confirm that inward deformations increase in early and late prophase, as well as upon calyculin A treatment, while it is unaffected by latrunculin A. Incidentally, we noticed that the typical shape of inward deformations fits well the theoretical shape of a membrane deformed by a localized pinning force[36], as reported in Fig. S7. These observations are in line with the notion that spindle-generated forces, rather than actomyosin contractility, are responsible for the development of severe, late-stage invaginations leading to NEBD[30]. On the other hand, membrane flickering is observed in late-G2, early prophase, and early calyculin A treatment, but not in latrunculin-A treated cells, leaving room for the hypothesis that it could arise from inner, chromatin-generated pulling forces.

**Chromatin density increases in correspondence to centripetal nuclear shape deformations.** To gain further insight into a possible role of chromatin in causing the observed centripetal shape fluctuations, we analyzed movies of HeLa cells double-transfected with H2B-mCherry and Emerin-GFP, as markers of chromatin and NE respectively. These experiments enabled us to monitor NE shape and chromatin density at the same time.

We first evaluated whether nuclei in early and late prophase could exhibit any separation between chromatin and lamina (as proxied by Emerin) at the site of invagination, and we observed no separation for the 11 cells analyzed in both cases (Fig. S8 shows an illustrative example). Subsequently, we quantified the cross-correlation between local deformations of NE and fluorescence intensity from histones in the corresponding area during invaginations (examples in Fig. S9, Video S7). We observed that in the neighborhood of invaginations inside the nucleus, while NE contour decreases at the angle of maximum inward pulling, the mean fluorescence intensity of chromatin increases. When there is an invagination, we also see the correlation, but not when there is not, as chromatin can condense locally without being attached to the NE. This observation clearly establishes a link between chromatin state and nuclear shape deformations in case of local reversible invaginations. In some cases (e.g. the second case reported in Fig. S9a), we saw that the chromatin signal increases a few seconds before the inward NE deformation reaches its maximum extent, suggesting that the local chromatin condensation leads to an increase of fluorescence that occurs before the NE invagination, and possibly pulling the NE itself inward. In Fig. S9b the Pearson correlation coefficient between NE deformations and chromatin fluorescence signal has been calculated for 10 sections of the NE with and without invaginations. This coefficient is close to -1 for invaginations, and 0 for non invaginations, supporting the correlation hypothesis. In brief, our data support a link between chromatin state and nuclear shape deformations at the site of local reversible invaginations.

To assess whether the cytoskeleton, and microtubules in particular, could also play a role in generating the transient invaginations observed in prophase, we transfected HeLa cells with Tubulin-mScarlet and Emerin-GFP to simultaneously monitor nuclear envelope (NE) fluctuations in relation to microtubule dynamics. As shown in Fig. S10, we observed that the transient inwards fluctuations we focused on in our study originated in NE areas far away from centrosomes and MT bundles, and independently of them (Video S8). On the other hand, we observed wide and deep irreversible invaginations leading to NE breakdown due to centrosome pushing at two poles of the nucleus (Video S9). We concluded that transient, reversible fluctuations were not related to centrosome pressure, while later severe, irreversible invaginations were at least in part the result of increasing centrosome pressure leading to NEBD, consistent with previous literature on mitotic progression[30].

**ATR inhibitors increase asymmetric deformations and effective bending modulus, and decrease effective tension.** ATR is a protein kinase belonging to the PI3KK family, mostly studied as a master regulator of the replication stress response and activator of the DNA damage checkpoint[37]. ATR has also been shown to respond to mechanical stimuli, mediating mechanotransduction at the nuclear envelope and aiding in chromatin release from the NE to alleviate topological stress[38,39].

Since chromatin tethering to the nuclear envelope is a crucial component of nuclear mechanics and a requirement for force propagation through the nucleus[40] and since ATR depletion leads to accumulation of long-lasting NE deformation[41], we tested whether ATR activity could also affect the NE fluctuations we observed in G2/M, by specifically inhibiting the kinase at mitotic entry. Interestingly, we found that ATR inhibition causes an increase in the number of deformations showing negative skewness (Fig. 3e). We could also observe a mild decrease in effective tension and an increase in effective bending modulus (Fig. S11, Video S10). The measured bending modulus increase in ATR-inhibited nuclei is a consequence of the drug having opposite effects on the small-$q$ (enhanced) and high-$q$ (depressed) part of the fluctuation spectrum, and might be the result of a previously reported change in the lipid composition of the NE upon ATR inhibition[41].

These results support the hypothesis that ATR inhibition correlates with longer-lasting centripetal invaginations, suggesting the coordinated process of chromatin release from the NE at early stages of mitosis is impaired whit ATR inhibition.

## Discussion

This study characterizes the nature of NE fluctuations preceding cell division in G2/M, confirming and expanding on previous findings[9,19]. Fig. S12 and Video S11 report kymographs and videos that summarize our major findings visually. By using different pharmacological perturbations, we could discern contributions from the cytoskeleton and chromatin in generating NE fluctuations, highlighting a role of chromatin in driving centripetal pulling motion. Our results suggest that nuclear shape fluctuations are driven by a combination of thermal motion and forces originating in the nucleus, while being affected by the surrounding cytoskeleton and its tethering with the NE. Accordingly, we found that latrunculin A (actin depolymerization) increases (symmetric) shape fluctuations (decreasing effective tension), and decreases nuclear radius, while blebbistatin (Myosin-II inhibition) increases nuclear radius and effective tension. These results suggest that the dynamic flickering of nuclear envelope might be constrained by the presence of actin stress fibers (which are lost with latrunculin A), possibly via LINC connections. On the other hand, the dynamic rearrangement of stress fibers caused by loss of myosin contractility has a more complex effective stiffening effect, which also leads to radius increase. This could be due to relaxation of a compressive constraint on the nucleus, and consequent change of the osmotic equilibrium towards a larger radius[12].

Once the genome has completed replication, the NE likely reaches a tenser status due to the increase in nuclear content[11,18]. We isolated events of localized transient invaginations that contribute to the asymmetric inwards fluctuations observed in late G2 and early prophase. This evidence supports active pinning centripetal forces that drive increasingly strong shape fluctuations (also resulting in a drop in effective tension) from G2 to mitosis, up until NE breakdown. Hence, (i) shape fluctuations can dramatically increase from G2 to mitosis, and (ii) they can become highly non-symmetric at this stage. Fluctuation asymmetry favoring inward displacements appears already in G2, together

with local reversible "pinning" centripetal deformations. These deformations become increasingly long lasting and irreversible as the cell cycle progresses towards NE breakdown, as contribution from spindle microtubules becomes increasingly prominent[30]. Microtubules, which play a pivotal role in the onset and progression of mitosis, do not appear to be directly involved in these reversible transient inwards fluctuations of the, but rather their contribution becomes increasingly predominant towards NE breakdown, through previously documented MTOCs-generated pushing forces that appear to be essential to initiate cell division[30]. Interestingly, latrunculin A (actin) and blebbistatin/Y27632 treatments (myosin) do not cause asymmetric shape fluctuations, while calyculin A (chromatin condensation) treatment makes them centripetal. Combining these findings (Table 1), we hypothesize a role of the NE as a dynamic sensor of cellular and nuclear mechanical stimuli. Chu and coworkers[9] interpreted the cell-cycle changes (G1-S-G2) as a change of material properties and/or of the forces driving the shape fluctuations. Our results support the idea that chromatin contributes to the forces driving these fluctuations through active pinning centripetal forces.

Loosening of chromatin state through TSA treatment at the onset of mitosis reduces the occurency of transient fluctuations, while simultaneous monitoring of microtubule dynamics suggest their generation is not primarily due to the cytoskeleton. These findings lead us to conclude that NE centripetal shape fluctuations can be mainly ascribed to chromatin dynamics. Starting from these data, it is possible to formulate some hypotheses on the force balance between physical processes regulating nuclear mechanics. Physically, nuclear size is mainly set by osmotic balance, and nuclear shape could be set by three mechanical components: chromatin, lamins, and the cytoskeleton. Chromatin and Lamin A are typically seen as resistive elements that together maintain nuclear shape. Lamins alone, on the other hand, cannot maintain nuclear shape, and the lamina buckles under mechanical stress when it is not supported by chromatin, suggesting a physical model of the nucleus as a polymeric shell enclosing a stiffer chromatin gel[42]. The cytoskeleton has varied and heterogeneous effects on nuclear structure, exerting active forces to stabilize it along the different stages of the cell cycle and in response to mechanical stimuli[28]. Disconnecting chromatin from the inner nuclear membrane results in softer nuclei that are deformable and more responsive to cytoskeletal forces[40], highlighting the importance of a balance between inward and outward forces at the nuclear boundary. Accordingly, recent studies reported the central role of nucleus-cytoskeleton interactions in processes such as nuclear positioning and stem cells differentiation[13,14].

Condensing chromatin exists in mechanically stressed states[43]. The idea that chromatin condensation could alter NE shape by exerting centripetal forces was suggested by previous observations on Drosophila salivary glands[44], where chromatin compaction forces were shown to drive distortions of the NE through chromatin-envelope interactions. Kumar and coworkers showed that chromatin-envelope interactions generate mechanical stress, which recruits and activates ATR kinase at the NE[38]. In line with their results, we observe increased negative skewness in nuclear shape deformations when ATR is inhibited just before entry in prophase. Additionally, a study on non-tumorigenic mammary epithelial cell MCF-10A has implicated chromatin in nuclear shape deformations, showing that these were independent of cytoskeletal connections[45]. Chromatin decompaction was further shown to cause nuclear blebbing, regardless of Lamin, as well as nuclear swelling[46,47]. Adding to these findings, our data suggest that the force balance at the NE is not static, and the nucleus progressing from S phase to G2 and mitosis feels increasing

**Table 1 Comparison of nuclear-shape fluctuation behavior before mitosis and under biochemical perturbations.**

| Perturbations | Biological effect | Fluctuations | Implications in shape fluctuations | Nucleus morphology | Refs. |
|---|---|---|---|---|---|
| Mitosis | Chromatin condensation | Increase (T&A) | Chromatin and cytoskeletal activity | More spherical | This study |
| Calyculin A | Induce chromatin condensation | Increase (A) | Chromatin involvement | Spherical and softer | This study |
| Latrunculin A | Inhibition of actin polymerization | Increase (A) | Cytoskeletal activity | Softer | This study[9], [9] |
| α-amanitin | Inhibition of polymerase II transcriptional activity | Increase (T&A) | Chromatin involvement | - | [9] |
| ATP depletion | Influence transcription, DNA replication, DNA repair, chromatin remodeling | Decrease (T&A) | Active (>4 s) and passive (>1s) fluctuations | - | [9] |
| Blebbistatin/ Y27632 | Inhibition of myosin II activity | Decrease (T&A) | Myosin II activity contribution | Slightly bigger | This study[9], [9] |
| Nocodazole | Inhibition of microtubule polymerization | Decrease (T&A) | Microtubules contribution | - | This study |
| ATR inhibitor | Reduced release of chromatin-envelope link | Increase (T&A) | Chromatin involvement | Invaginations micronuclei | This study[41], |

Legend: T = thermally driven and A = active.
Data from Figs. 1 and 2, and ref. [9].

extensile stress, and increasing localized stress from inner chromatin, affecting its shape fluctuations. Isotropic contributions to these stresses also likely come from forces of osmotic origin[11,48].

Based on our experiments, we formulate the hypothesis of a nucleus under stabilizing stress from osmotic balance and the external cytoskeleton, prone to experience local inward pulling forces coming from the underlying condensing chromatin. Calyculin treatment, and fast joint video acquisition of H2B histones, Emerin and Tubulin, lead us to conclude that these local pinning forces (becoming more widespread as G2 progresses to mitosis) may come from condensing chromatin. This adds a component to the widely accepted view of chromatin as a gel conferring structural integrity and stiffness to the nucleus[42], because locally condensed chromatin induces active forces that can destabilize the nuclear structure. The decrease in effective tension under latrunculin A treatment may be compatible with the idea of an extensile stress applied by the cytoskeleton. While the cytoskeleton is typically pictured as a compressive force, the perinuclear actin cap has been shown to stabilize nuclear shape[49]; disconnecting chromatin from the inner nuclear membrane results in softer nuclei that are deformable and more responsive to cytoskeletal forces[40]. A recent study has shown that chromatin status determines its interaction with the mitotic spindle, highlighting the importance of the nucleus physical properties for correct mitotic progression[35]. When considered alongside our results, these findings suggest that the global and local balance of forces exerted and experienced by the nucleus are crucial for ensuring proper genome segregation, and may result in delays or abortion of mitotic progression when impaired by physical or genetic interference.

The phenomena reported here likely play a role in the coordination of chromatin condensation and NE breakdown during mitosis, in a similar way as NE reassembly is coordinated with chromosome segregation[50]. In line with recent findings reporting the requirement of modification in NE tension for nuclear internalization of cyclin B and progression through mitosis, the shape fluctuations reported here may be not only a byproduct, but also a driver of cell-cycle progression. Since chromatin pulling events deforming the nucleus develop into widespread invaginations whose instability eventually corresponds to NEBD, we speculate that the intensity of the opposed forces on the NE increases during G2 and mitosis, and may work in parallel with spindle-generated forces directing NE breakdown[30]. Centripetal pulling by chromatin could mechanically induce a rupture-prone state in the NE before breakdown, facilitate internalization of mitotic regulators[19] or trigger mechanosensitive signaling cascades, as in the case of the cPLA2 protein[18], ultimately leading to cell commitment to division.

## Methods

**Cell culture, plasmids, and transfection.** HeLa cells stably expressing m-Cherry-H2B (reported by Kumar and coworkers[38]) were maintained in DMEM (Dulbecco's Modified Eagle's medium) with GlutaMAX (Life Technologies) supplemented with 10% (vol/vol) fetal bovine serum (FBS, Biowest), and penicillin-streptomycin (Microtech), in a humidified incubator atmosphere at 37 °C and 5% CO2. Lipofectamine2000 (Invitrogen) was used for transfecting plasmids Emerin pEGFP-C1 (637) plasmid (Addgene ID-61993) and pmScarlet-i_alphaTubulin_C1 (Addgene ID-85047) into cells, using the protocol recommended by the manufacturer. The following day, cells were plated onto fibronectin-coated glass coverslips (10 µg/ml; 30 min; at 37 °C). Experiments were performed ~36–48 h after transfection.

**Cell-cycle phase determination.** Cells were arrested at the G1/S transition (by a double thymidine block) or at the G2/M transition (by CDK1 inhibition) and then released into S phase or mitosis respectively, by washing three times with 1x PBS before adding fresh medium. Analysis of mean cell cycle duration of HeLa cells by time-lapse videos showed that HeLa cells spent on average 16.48 h in interphase and 1.75 h undergoing mitosis (Fig. S1a). Reported timings of HeLa cell cycle stages duration are 7–8 h for G1 phase, 7–8 h for S phase, and 2–3 h for G2[22]. We

used video microscopy to confirm that overall time spent by our HeLa cells in interphase and mitosis was fully in line with published data (Fig. S1a). Early G1 was defined as [0,4] h after mitosis release, late G1 was defined as [4,8] hours after release from mitosis, early S phase was defined as [0,3] hours post G1/S release from thymidine block, late S phase was defined as [4,7] hours post G1/S release, and G2 phase cells were obtained after 16 h of CDK1 inhibitor treatment (Fig. 1a). Early and late prophase were determined respectively as 10 min and 30 min after release from CDK1 inhibition.

**Drug treatments**. For cell synchronization in the G2-M transition, first cells were treated with thymidine (2 mM-Sigma Aldrich) for 14 h, washed with PBS, released for 7 h and then incubated further for 16 h with Cdk1 inhibitor, RO-3306 (Seleckchem-S7747) at 10 μM concentration. For cell synchronization in the G1-S transition, cells were treated with thymidine (2 mM-Sigma Aldrich) for 18 h, washed with PBS, released for 9 h and then again treated with Thymidine for 18 h. Calyculin A (Cell signaling technology -9902s) and latrunculin A (Sigma Aldrich-428021) were commercially purchased. Calyculin A was used at 5 nM and latrunculin A was used at 1 μM concentration. Inhibitors were added to the media during the experiment, after pre-treatments acquisitions, and were maintained throughout the course of the experiment. Post calyculin A treatment, we divided the cells into two groups, named "early" and "late" phases based on their progress in rounding up and subsequent radius decrease. Cells with nuclear contour resembling that of pre-treatment are called early phase calyculin A. Cells with significantly lower nuclear radius (at least 10% less than pre-treatment) and complete deformed contour are defined late stage calyculin A. These choices are supported by time-lapse videos where the full development of the drug effect is visible (Video S12). Cells becoming rounder during the acquisition were not considered for further analysis. Latrunculin A was added to cells incubated for 16 h with Cdk1 inhibitor (RO-3306). Videos were acquired for about 140s (see below), 20 and 50 min after the treatment. Mild increase in radius of G2-arrested cells compared to the regular G2 (from cell cycle analysis) is due to their prolonged arrest in G2. For Rho-associated protein kinase (ROCK) inhibition, 10 μM Y27632 inhibitor was administered to cells for 30 min prior to image acquisition (Video S6). For treatments with blebbistatin, cells were treated with blebbistatin (Sigma Aldrich) at 5 μM concentration for 45 min inside a dark incubator chamber to avoid photo-inactivation of the drug, then imaged for 140 s. For double treatments with blebbistatin and calyculin A, cells inside a dark incubator chamber were first treated with blebbistatin (Sigma Aldrich) at 5 μM concentration for 30 min and then treated with calyculin A (15 min) in presence of blebbistatin, and subsequently imaged for 140 s. For trichostatin A treatment (Fig. S6), cells were incubated with 5 μM TSA (Tocris-1406) for 2 h before RO3306 washout. For ATR inhibition experiments (Video S10), 1 μM of ATR inhibitor VE822 was added 2 h prior to release from RO-3306. Cells were kept in the same inhibitor concentration throughout mitotic progression.

**Cell lysis and Immunoblotting**. Cells were lysed with lysis buffer (50 mM Tris-HCl pH 8.0, 1 mM MgCl2, 200 mM NaCl, 10% Glycerol, 1% NP-40) supplemented with protease (Roche) and phosphatase inhibitors (Sigma). Cell lysates boiled with Laemmli buffer were resolved on Mini-PROTEAN® (Biorad) precast gels, transferred to 0.45 nitrocellulose membrane, and probed overnight at 4 °C with primary antibodies against pospho-Lamin A/C (Ser22) (D2B2E from CST) and vinculin (V9131 from Sigma Aldrich). After washings with 1X PBS, membranes were incubated with secondary antibodies for 1 h at RT and acquired using ChemiDoc imaging system (Image Lab v5.0).

Exponentially growing HeLa cells were treated with either 5 nM calyculin A or 5 nM DMSO for 20 minutes prior to cell harvest in 200 μl of SDS lysis buffer containing 1 M DTT. Samples were boiled for 10 min at 95 °C then separated by polyacrylamide gel electrophoresis in the presence of SDS and transferred onto 0.45 nitrocellulose membranes. Membranes were blocked with 5% milk powder in Tris-buffered saline containing 0.05% Tween (TBS-Tween) for 1 h at room temperature, then primary antibodies were added in 5% milk powder in TBS-T and incubated over night at 4 °C. Membranes were washed 3 times with TBST then incubated with secondary anti-mouse and anti-rabbit antibodies for 1 h at room temperature. After three washings with TBST, membranes were rinsed with SuperSignal™ West Femto Maximum Sensitivity Substrate and developed at a ChemiDoc Imaging System (Bio-rad).

**Imaging and image processing**. Confocal Spinning Disk microscope (Olympus) equipped with IX83 inverted microscope provided with an IXON 897 Ultra camera (Andor), Software cellSens Dimension 1.18, and attached with ×100 silicone immersion objective (Refractive Index = 1.406; Numerical Aperture = 1.35) was used for HeLa cell imaging. 500 frames were acquired sequentially from Green (488 nm) and red (561 nm) channels at a maximum speed with individual exposure time of 100 ms (~4 frames per second). For cell cycle based analysis, time-points were taken every 3–4-h interval by acquiring 500 frames of each channel. Each cell was imaged for maximum of 5 time-points in a span of 12 h and long-term acquisitions from the same cell was avoided to reduce the effect of phototoxicity. Cells were synchronized and released to univocally assign their cell cycle stage by monitoring their growth along the 12 h. For treatments, multiple position

acquisition was used to acquire the same cells at different time-points. Images were then processed using the ImageJ software.

Effective bending modulus and tension of the NE were obtained by fitting the fluctuation spectrum with Eq. (1) for modes 6-34. Modes below 5 were excluded because influenced by the cell shape[24] and higher modes above 34 were affected by noise due to the acquisition exposure time. From fluctuation dynamics, relaxation time of mode 3 was obtained by fitting the autocorrelation function of the fluctuation amplitudes for mode 3 with a single exponential[3]. Invaginations were first identified by change in contour fluctuations and confirmed by looking at videos. The mean of the contour shape for the first 10 frames was subtracted from the contour of each frame as reference. The depth is the difference between the steady state contour and the minimum of the invagination. The width is determined by the points corresponding to 10% of the depth.

**Statistics and reproducibility**. All the statistics and the numbers of tecnical and biological replicates are in the Table S1–S3 and their captions.

**Convention for Fourier transform in the flickering code**. Equation (1) is derived in ref. [3] and uses the following non-unitary convention for the 2D Fourier transform of the displacement function $u(\vec{x})$:

$$u(\vec{x}) = \frac{A}{(2\pi)^2} \int d\vec{q}\, u_{\vec{q}}\, e^{i\vec{q}\,\vec{x}} \tag{2}$$

and the inverse transform is

$$u(\vec{q}) = \frac{1}{A} \int d\vec{x}\, u_{\vec{x}}\, e^{-i\vec{q}\,\vec{x}} \tag{3}$$

where $A = L \times L$ is the area of the membrane.

In order to match the Fourier transform with the discrete Fourier series calculated in Matlab (Fast Fourier Transform, FFT), the Fourier coefficients coming out from Matlab's FFT of $u$ need to be corrected by:

$$u_q = h_q^{Matlab} \times \frac{\Delta x}{L} \tag{4}$$

$$i.e. h_q^2 = h_q^{Matlab} \times \left(\frac{1}{N}\right)^2. \tag{5}$$

**Reporting summary**. Further information on research design is available in the Nature Portfolio Reporting Summary linked to this article.

## Data availability
All data supporting this study are provided as electronic supplementary material accompanying this manuscript and are available in the public Mendeley repository: https://doi.org/10.17632/pwwpt8s9cb.1[51]. All other data are available from the corresponding author on reasonable request. Uncropped and unedited blot/gel images are provided as Fig. S13.

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

## Acknowledgements

We are grateful to Marco Foiani for sharing personnel and resources from his Laboratory to carry out this project. The authors also thank Paolo Maiuri, Giorgio Scita, Nishit Srivastava and Matthieu Piel for useful feedback on this manuscript, and Nils Gauthier for sharing reagents. M.C.L. was supported by the Italian Association for Cancer Research (AIRC), grant AIRC-IG (REF: 23258). P.C. was supported by the Engineering and Physical Sciences Research Council (EPSRC) (EP/R011443/1). Work of G.R.K. and G.P. was supported by Italian Association for Cancer Research (AIRC) AIRC-IG REF: 21416, PI Marco Foiani. V.I. was funded by the EPSRC and Sackler scholarships, and by the Wellcome Trust Junior Interdisciplinary fellowship (Wellcome 20485/Z/16/Z). G.R.K. was supported by Marie Curie Initial Training Networks (ITN), (FP7 -aDDRess') fellowship and Italian Association for Cancer Research (AIRC) fellowship.

## Author contributions

M.C.L., G.R.K., and P.C. conceived research, and together with V.I. designed research; G.R.K. and G.P. performed experiments; V.I. analyzed data; G.R.K., V.I., G.P., and M.C.L. wrote the manuscript; P.C. contributed in reviewing and editing the manuscript.

## Competing interests

The authors declare no competing interests.
