## [Peer Review File · Communications Biology]

Reviewers' comments:

Reviewer #1 (Remarks to the Author):

Authors analyze the contribution of both the cytoskeleton and chromatin in the shape fluctuations of the nucleus during the cell cycle of HeLa cells. They first show that nuclear shape fluctuations vary during the cell cycle. Using various drug treatments, they propose that both the cytoskeleton and chromatin condensation contribute to changes in nuclear shape. The study investigates the contribution of an important player here, namely chromatin condensation in nuclear shape fluctuations, however the non-specific nature of the tools used to tackle this question (mainly drug treatment) do not allow to make solid conclusions.

Comments:

1/ to test the influence of chromatin condensation on nuclear shape fluctuations, authors use Calyculin A, a phosphatase inhibitor, that is clearly not very specific, yet often used in many works. It would be nice if authors could use another tool to test for the influence of chromatin condensation on nuclear shape. Maybe a condensing mutant?

Indeed, previous work using genetic tools has shown, that despite a modest difference in chromatin condensation, the cytoplasmic actin cytoskeleton in cell physiologically arrested in G2 phase of the cell cycle, mouse oocytes, is a primary contributor of nuclear shape and of nuclear envelope fluctuations (Almonacid Dev Cell 2019), arguing that chromatin plays a minor role.

2/ It would be more accurate for authors to measure the changes in nuclear volume rather than nuclear area in Fig 2b. This measure would provide a more precise estimate of nuclear tension (Fig 2c).

3/ Authors describe typical inward deformations and interpret them as regions of chromatin condensation. Notably in their discussion (p19) they mention that the fluctuations become highly asymmetric. Authors should look at Almonacid Dev Cell 2019 and at Biedzinski EMBO J 2020, both papers which clearly showed the importance of MTOCs and microtubules in promoting nuclear envelope fluctuations. Maybe authors should label microtubules and centrosome in live, to determine if in their HeLa cells the major inward deformations correspond to region of increased nucleation of microtubules due to centrosome presence?

Reviewer #2 (Remarks to the Author):

The work by Introini et al. presents an interesting set of experimental data and theoretical considerations revealing a hitherto unappreciated role of nuclear chromatin compaction as an internal driver of nuclear envelope (NE) instability. In order to strengthen the conclusions of the paper, I propose the following additional experiments:

1) The use of Calyculin as the only perturbation tool (despite all the additional controls) is the main weakness of the story; therefore, I'd try to perform measurements of NE fluctuations in cells treated with more specific chromatin-modifying drugs and metabolites (e.g., EZH2 inhibitors, HDAC inhibitors, etc.).

2) In order for chromatin compaction to trigger NE invagination, chromatin has to be attached to the NE. What will happen to NE fluctuations if the authors experimentally ablate linkages between the NE and chromatin (e.g. via overexpression of dominant-negative mutants of LBR, or BANF1, etc.)?

3) Sorry if I missed it throughout the text, but do the authors have any experimental and/or theoretical data on what will happen to the process of NE breakdown (NEBD) and progression towards mitosis if the contribution of chromatin compaction is entirely excluded from the process? Will cells undergo NEBD? If so, how fast/efficient? I speculate here that NEBD will take place but might go slower.

Fast vs. slow transitions through different stages of the cell cycle are observed in rapidly proliferating human epithelial stem cells at the edge of the wound vs. the bulk of tissue. At the same time, changes in chromatin organization (e.g. compaction) are a fundamental attribute of stemness and pluripotency in various stem cell systems. I am curious can the work by Introini et al. help better understand human stem cell biology with regards to the unique chromatin state of these cells and their ability to proliferate fast? Adding this or a similar angle to the story by Introini et al. will significantly boost its impact.

Dear Referees,

We are grateful for the work performed on our manuscript.

We have substantially revised our work addressing the comments and suggestions. In particular, **(i) we are now able to provide an orthogonal method to chromosome condensation via Caliculyn A**, by altering chromosome condensation **via HDAC inhibitors**, obtaining experimental results that are in line with our hypotheses; **(ii) we can prove that the localized transient invaginations** that we observe in early mitosis **are not originated from MTOC pushing**, by testing the contribution of centrosome positioning with additional experiments.

We report below our point-by-point answers to your comments. Reviewer remarks are in red and responses are in black.

Kindest Regards,

Marco Cosentino Lagomarsino (on behalf of the authors)

Reviewers' comments:

Reviewer #1 (Remarks to the Author):

Authors analyze the contribution of both the cytoskeleton and chromatin in the shape fluctuations of the nucleus during the cell cycle of HeLa cells. They first show that nuclear shape fluctuations vary during the cell cycle. Using various drug treatments, they propose that both the cytoskeleton and chromatin condensation contribute to changes in nuclear shape. The study investigates the contribution of an important player here, namely chromatin condensation in nuclear shape fluctuations, however the non-specific nature of the tools used to tackle this question (mainly drug treatment) do not allow to make solid conclusions.

We appreciate the constructive comments provided by the reviewer. We acknowledge the limitations of our study and agree with the suggestions made by the reviewer. To address these limitations, we have conducted and included two new key experiments that significantly strengthen our findings: **chromosome condensation via HDAC inhibitors and microtubule effect via HeLa cells transfected with tubulin-mScarlet.**

There remain some open questions, which we are actively addressing in our labs. However, as these questions pertain to broad and specific topics such as chromosome segregation and spindle mechanics, we believe that they require a more in-depth characterization through independent studies beyond the scope of the present manuscript.

We have also revised our conclusions to accurately reflect our findings. We believe that our results provide sufficient evidence to support the interpretation that (i) chromatin condensation is associated with increased nuclear fluctuations (particularly localized transient invaginations) which tend to be inward-directed, and (ii) while the perinuclear cytoskeleton may play a role in this phenomenon, it is not the sole explanation.

As pointed out by the reviewer, we believe that our findings have relevant implications in our understanding of chromatin condensation and nuclear fluctuations. Since our findings suggest that chromatin condensation is coupled with (some, not all) nuclear fluctuations, this in turn suggests that other factors, such as changes in histone modifications (some of which we tested in this revision, see below), may also play a role, warranting further investigation.

Comments:

1/ to test the influence of chromatin condensation on nuclear shape fluctuations, authors use Calyculin A, a phosphatase inhibitor, that is clearly not very specific, yet often used in many works. It would be nice if authors could use another tool to test for the influence of chromatin condensation on nuclear shape. Maybe a condensing mutant?

Indeed, previous work using genetic tools has shown, that despite a modest difference in chromatin condensation, the cytoplasmic actin cytoskeleton in cell physiologically arrested in G2 phase of the cell cycle, mouse oocytes, is a primary contributor of nuclear shape and of nuclear envelope fluctuations (Almonacid Dev Cell 2019), arguing that chromatin plays a minor role.

As mentioned by the reviewer, Calyculin A, similar to other common chromatin-condensing drugs such as Okadaic acid, is a broad inhibitor of phosphatases, yet often used because chromatin condensation is a finely regulated process which is hard to isolate and target specifically.

Following the reviewer's suggestion, we included another independent perturbation whose effect could be compared to chromatin state in prophase. We performed additional experiments treating cells with Trichostatin A (TSA), a pan-histone deacetylase inhibitor, as previously described in Schneider *et al.* Nature 2022. We reasoned that a looser and less compact chromatin would result in reduced inward pulling and a lower amount of chromatin-driven NE invaginations. As we show in **Supplementary Figure S6**, TSA-treated cells show a significantly reduced number of transient invaginations compared to control cells, as opposed to highly fluctuating nuclei in Calyculin A treated cells. These results support the hypothesis that the chromatin state directly affects NE fluctuations, by investigating two conditions that lead to opposite outcomes: reduced local invaginations with TSA treatment or increased fluctuations due to inward pulling via Caliculyn A condensation.

Additionally, in the **Discussion** section of the revised manuscript, we comment on the study by Almonacid and co-workers, together with the importance of genetic perturbations in past and for future studies. The analysis we performed of chromatin-driven forces inducing NE fluctuations does not exclude the well-recognised role of cytoskeletal structures in regulating nuclear mechanotransduction, but it complements this picture by highlighting the role of nuclear mechanics on the inner side of the envelope.

2/ It would more accurate for authors to measure the changes in nuclear volume rather than nuclear area in Fig 2b. This measure would provide a more precise estimate of nuclear tension (Fig 2c).

The fast acquisition rate required to capture the membrane fluctuation spectrum (approximately 1 image/0.25 s) poses a technical limitation to z-stack acquisition. The z-axis resolution is low, thus nuclear median profile is the best proxy for nuclear membrane. This method is the state-of-the-art for flickering spectroscopy techniques, and it has been extensively used to recover tension from shape fluctuations in vesicles, red blood cells and nuclei. The nucleus is more complex, and no technique can currently really determine an absolute tension. Analysis of changing nuclear volume could relate to membrane tension but would also be sensitive to other processes that change the volume. An advantage of flickering is that although we do not know the scale of energy (these are not thermal fluctuations) and hence we can only describe effective tension, we are actually able to show from the shape of the fluctuation spectrum that we are probing a quantity that is proportional specifically to tension. Hence we think NE fluctuations, by imaging the equatorial section of the cell nucleus as marked by the emerlin-GFP signal, are currently the most insightful measurement.

3/ Authors describe typical inward deformations and interpret them as regions of chromatin condensation. Notably in their discussion (p19) they mention that the fluctuations become highly asymmetric. Authors should look at Almonacid Dev Cell 2019 and at Biedzinski EMBO J 2020, both papers which clearly showed the importance of MTOCs and microtubules in promoting nuclear envelope fluctuations. Maybe authors should label microtubules and centrosome in live, to determine if in their HeLa cells the major inward deformations correspond to region of increased nucleation of microtubules due to centrosome presence?

The reviewer correctly highlights the importance of microtubules and the cytoskeleton in contributing to force transduction to the nucleus. It is important to state that it was never our intention to state that chromatin condensation would be the only player. We have now made sure that our **Introduction** refers to this body of knowledge.

To directly investigate whether the transient invaginations we observed were mainly caused by microtubules (MTs), we followed the reviewer's suggestion and conducted additional experiments by transfecting HeLa cells with tubulin-mScarlet and emerin-GFP to simultaneously monitor nuclear envelope (NE) fluctuations in relation to MT dynamics.

As shown in **Supplementary Figure S10**, we observed that the transient inwards fluctuations we focused on in our study originated in NE areas far away from centrosomes and MT bundles, and independently of them (**Supplementary Video S8**). On the other hand, deep, irreversible invaginations leading to NE breakdown were observed due to centrosome pushing at two poles of the nucleus (**Supplementary Video S9**). When combined with the observation that chromatin condensation occurred at sites of NE oscillation, we concluded that transient, reversible fluctuations were mainly driven by chromatin inward forces, while severe, irreversible invaginations were the result of increasing centrosome pressure leading to NEBD, consistent with previous literature on mitotic progression.

As the reviewer pointed out, previous studies (e.g., Almonacid *et al.*, Biedzinski *et al.*, and some of the studies cited in our work, such as Beaudouin *et al. Cell 2002*) have reported the crucial role played by the cytoskeleton in controlling nuclear shape throughout different stages of the cell cycle and in different cell models. While major deformation events related to oocytes nuclear positioning, stem cell differentiation, and NEBD as reported in the aforementioned papers arise from a predominant contribution from the cytoskeleton, our study focused on transient, reversible shape fluctuations occurring along the NE perimeter and found a direct role of chromatin dynamics in their regulation.

Reviewer #2 (Remarks to the Author):

The work by Introini et al. presents an interesting set of experimental data and theoretical considerations revealing a hitherto unappreciated role of nuclear chromatin compaction as an internal driver of nuclear envelope (NE) instability. In order to strengthen the conclusions of the paper, I propose the following additional experiments:

- 1) The use of Calyculin as the only perturbation tool (despite all the additional controls) is the main weakness of the story; therefore, I'd try to perform measurements of NE fluctuations in cells treated with more specific chromatin-modifying drugs and metabolites (e.g., EZH2 inhibitors, HDAC inhibitors, etc.).*

Following the reviewer's suggestion (and a similar point raised by Reviewer #1), we treated cells with Trichostatin A (TSA), an HDAC inhibitor, as an additional perturbation to compare with calyculin A treatment and physiological prophase condensation. As shown in **Supplementary Figure S6**, we show that following chromatin decompaction through TSA treatment, the overall fluctuation spectrum does not significantly change from the control condition, while the number of localized reversible invaginations is strongly reduced. This reinforces our interpretation that chromatin-driven forces are important players in generating transient NE invaginations.

2) In order for chromatin compaction to trigger NE invagination, chromatin has to be attached to the NE. What will happen to NE fluctuations if the authors experimentally ablate linkages between the NE and chromatin (e.g. via overexpression of dominant-negative mutants of LBR, or BANF1, etc.)?

We agree with the reviewer that chromatin tethering to the NE is indeed very likely to be a necessary condition for forces generated inside the nucleus to propagate to its surface, generating the observed fluctuations. We addressed the effect of interfering with chromatin tethering by inhibiting ATR specifically at the G2/M boundary (**Supplementary Figure S11**). ATR was previously shown to interact with the NE in prophase and to mediate the release of topological stress at the chromatin-envelope interface (Kumar et al. Cell 2014, Bermejo et al. Cell 2011). As expected, inhibiting ATR prior to mitotic entry results in increased NE fluctuations, indicating that a more persistent tethering enhances the force load experienced by the envelope. This indirectly suggests that lack of chromatin-envelope tethering could result in lower membrane fluctuations, eventually affecting the following progression in mitosis. As suggested by the reviewer, we also adjusted the manuscript section reporting the ATR inhibition results, to better reflect its implication on chromatin tethering to the NE.

3) Sorry if I missed it throughout the text, but do the authors have any experimental and/or theoretical data on what will happen to the process of NE breakdown (NEBD) and progression towards mitosis if the contribution of chromatin compaction is entirely excluded from the process? Will cells undergo NEBD? If so, how fast/efficient? I speculate here that NEBD will take place but might go slower.

In our study, we focused on nuclear membrane fluctuations that occur before NEBD and we did not explore how the absence of these fluctuations could affect mitotic progression, although we agree with the reviewer that it is an interesting question. The effects of interfering with the biophysical properties of chromatin and the outcomes of cell division are currently being investigated in parallel ongoing studies by the Foiani group.

In a recent study by Schneider et al. published in Nature 2022, the physical state of chromatin was found to play a critical role in determining its interaction with the spindle and whether it is penetrable or impenetrable to microtubules, depending on the mitotic stage. In this study, the authors treated cells with TSA, which induced a change in the state of chromatin that made it more accessible to microtubules, preventing proper attachments that ensure segregation. When considered alongside our results, these findings suggest that the balance of forces exerted and experienced by chromatin is crucial for ensuring proper segregation of the

genome during mitosis and may result in delays or abortion of mitotic progression, as hypothesized by the reviewer.

We have now added these considerations to our **Discussion** section.

Fast vs. slow transitions through different stages of the cell cycle are observed in rapidly proliferating human epithelial stem cells at the edge of the wound vs. the bulk of tissue. At the same time, changes in chromatin organization (e.g. compaction) are a fundamental attribute of stemness and pluripotency in various stem cell systems. I am curious can the work by Introini et al. help better understand human stem cell biology with regards to the unique chromatin state of these cells and their ability to proliferate fast? Adding this or a similar angle to the story by Introini et al. will significantly boost its impact.

We are grateful for the reviewer's valuable advice on other potential implications of our study that we did not consider in the manuscript. Various studies have investigated the effect of force transmission to and from chromatin as a critical determinant of cell fate, particularly in stem cells. Although our study did not generate enough data to prove any potential consequences of disrupted force balance resulting in altered membrane fluctuations, it is possible that varying degrees of flexibility or deformability of the NE are required depending on the proliferative demand of different cell types. A systematic analysis of nuclear shape fluctuations with our technique in different physio-pathological models could provide new insights into the mechanisms regulating cell fate through nuclear mechanotransduction. We have included a sentence in our **Discussion** to address this point.

REVIEWERS' COMMENTS:

Reviewer #1 (Remarks to the Author):

Authors have answered all my comments, hence I am fully satisfied with the revised version of their manuscript.

Reviewer #2 (Remarks to the Author):

In my opinion, the authors were able to address all the major critical points. Of course, like with any other paper, there is still room for improvement. But I would not delay the publication of this work by requesting additional experiments or significant changes. I believe it is important to report these new data in a timely fashion. The work supports an exciting new idea that nuclear forces can be generated "inside-out" via regulated changes in the physicochemical properties of nuclear chromatin.

One minor suggestion:

If possible, please add this reference
<https://pubmed.ncbi.nlm.nih.gov/30633879/> to the Introduction section (page 3).